# SoK: Analysis Techniques for WebAssembly

Håkon Harnes * and Donn Morrison *

Department of Computer Science, Norwegian University of Science and Technology, 7034 Trondheim, Norway
* Correspondence: haakaha@alumni.ntnu.no (H.H.); donn.morrison@ntnu.no (D.M.)

**Abstract:** WebAssembly is a low-level bytecode language that enables high-level languages like C, C++, and Rust to be executed in the browser at near-native performance. In recent years, WebAssembly has gained widespread adoption and is now natively supported by all modern browsers. Despite its benefits, WebAssembly has introduced significant security challenges, primarily due to vulnerabilities inherited from memory-unsafe source languages. Moreover, the use of WebAssembly extends beyond traditional web applications to smart contracts on blockchain platforms, where vulnerabilities have led to significant financial losses. WebAssembly has also been used for malicious purposes, like cryptojacking, where website visitors' hardware resources are used for crypto mining without their consent. To address these issues, several analysis techniques for WebAssembly binaries have been proposed. This paper presents a systematic review of these analysis techniques, focusing on vulnerability analysis, cryptojacking detection, and smart contract security. The analysis techniques are categorized into static, dynamic, and hybrid methods, evaluating their strengths and weaknesses based on quantitative data. Our findings reveal that static techniques are efficient but may struggle with complex binaries, while dynamic techniques offer better detection at the cost of increased overhead. Hybrid approaches, which merge the strengths of static and dynamic methods, are not extensively used in the literature and emerge as a promising direction for future research. Lastly, this paper identifies potential future research directions based on the state of the current literature.

**Keywords:** WebAssembly; vulnerability analysis; browser security; cryptojacking; smart contracts

## 1. Introduction

The Internet has come a long way since its inception and one of the key technologies that have enabled its growth and evolution is JavaScript. JavaScript, which was developed in the mid-1990s, is a programming language that is widely used to create interactive and dynamic websites. It was initially designed to enable basic interactivity on web pages, such as form validation and image slideshows. However, it has evolved into a versatile language that is used to build complex web applications. Today, JavaScript is one of the most popular programming languages in the world, currently being used by 98% of all websites [1].

Despite its popularity and versatility, JavaScript has some inherent limitations that have become apparent as web applications have grown more complex and resource-demanding. Specifically, JavaScript is a high-level, interpreted, dynamically typed language, which fundamentally limits its performance. Consequently, it is not suited for developing resource-demanding web applications. To address the shortcomings of JavaScript, several technologies, like ActiveX [2], NaCl [3], and asm.js [4], have been developed. However, these technologies have faced compatibility issues, security vulnerabilities, and performance limitations.

WebAssembly was developed by a consortium of companies, including Mozilla, Microsoft, Apple, and Google, as a solution to the limitations of existing technologies. WebAssembly is designed as a safe, fast, and portable compilation target for high-level

languages like C, C++, and Rust, allowing them to be executed with near-native performance in the browser. It has gained widespread adoption and is currently supported by 96% of all browser instances [5]. Moreover, WebAssembly is also being extended to desktop applications [6], mobile devices [7], cloud computing [8], blockchain virtual machines (VMs) [9–11], IoT [12,13], and embedded devices [14].

However, WebAssembly is not without its own set of challenges. Vulnerabilities in memory-unsafe languages, like C and C++, can translate into vulnerabilities in WebAssembly binaries [15]. Unfortunately, two-thirds of WebAssembly binaries are compiled from memory-unsafe languages [16], and these attacks have been found to be practical in real-world scenarios [15]. Vulnerabilities have also been uncovered in WebAssembly smart contracts [17,18], consequently causing significant financial loss. Moreover, WebAssembly has been used for malicious purposes, such as cryptojacking, where the hardware resources of website visitors are used for crypto mining without their consent [19]. To mitigate these issues, several analysis techniques for WebAssembly binaries have been proposed.

In this paper, we conduct an in-depth literature review of analysis techniques for WebAssembly binaries, with a focus on their application across diverse computing environments, including web development, cloud computing, and edge computing. To this end, we classify the analysis techniques based on their strategy and objectives, uncovering three primary categories: detecting malicious WebAssembly binaries (Section) 5.1), detecting vulnerabilities in WebAssembly binaries (Section 5.2), and detecting vulnerabilities in WebAssembly smart contracts (Section 5.3). Moreover, we compare and evaluate the techniques using quantitative data, highlighting their strengths and weaknesses. Lastly, one of the main contributions of this paper is the identification of future research directions based on the literature review conducted. In summary, this paper contributes the following:

- A comprehensive analysis of current analysis techniques for WebAssembly binaries, using quantitative data to evaluate their strengths and weaknesses.
- A taxonomical classification of current analysis techniques for WebAssembly binaries.
- Key findings and limitations of current analysis techniques for WebAssembly binaries, including the trade-offs between accuracy and overhead of static and dynamic analysis methods.
- Identification of gaps in the literature and suggestions for future research directions.

The rest of this paper is structured as follows: Section 2 provides the necessary background information and the current state of research in the field. Section 3 reviews related work, highlighting previous studies and their contributions. Section 4 describes the methodology employed in our research, including the search strategy, selection process, data extraction, and analysis methods. The main findings of the systematic review are detailed in Section 5, where we categorize and evaluate the analysis techniques for WebAssembly based on their method and effectiveness. A discussion of these findings and their implications are presented in Section 6. Finally, Section 7 concludes the paper and suggests future research directions.

## 2. Background

The background section of this paper provides a detailed overview of WebAssembly. The limitations of JavaScript and prior attempts at incorporating low-level code on the web are first discussed. Then, an in-depth description of WebAssembly's security mechanisms, vulnerabilities, and use cases are presented.

### 2.1. History

**JavaScript**. Initially, the Internet was primarily used by researchers, scientists, and other academics to share information and collaborate on projects. At this time, websites were mostly composed of static text and images, lacking dynamic or interactive components. The arrival of web browsers such as Netscape Navigator and Internet Explorer in the late 1990s made the Internet accessible to the general public and sparked the development of technology to enhance website user experience with dynamic and interactive elements.

JavaScript, created by Netscape in 1995 [20], became one of these technologies, enabling web developers to create engaging content. Today, JavaScript is a widely used programming language supported by all major web browsers and used on 98% of websites [1].

Despite its popularity and versatility, JavaScript has some inherent limitations that impact its performance. As a high-level language, JavaScript abstracts away many of the details of the underlying hardware, making it easier to write and understand. However, this also means that the JavaScript engine has to do more work to translate the code into machine-readable instructions. Additionally, because JavaScript is an interpreted language, it must be parsed and interpreted every time it is executed, which can add overhead and decrease performance. Lastly, JavaScript is dynamically typed, meaning that the type of a variable is determined at runtime. This can make it difficult for the JavaScript engine to optimize the code, resulting in reduced performance. These limitations can hinder the performance of JavaScript in resource-demanding or complex applications. There is, therefore, a need for high-performance, low-level code on the web.

**ActiveX**. ActiveX [2] is a deprecated framework that was introduced by Microsoft in 1996. It allowed developers to embed signed x86 binaries through ActiveX controls. These controls were built using the Component Object Model (COM) specification, which was intended to make the controls platform-independent. However, ActiveX controls contain compiled x86 machine code and calls to the standard Win32 API, restricting them to x86-based Windows machines. Additionally, they were not run in a sandboxed environment, consequently allowing them to access and modify system resources. In terms of security, ActiveX did not ensure safety through its technical design but rather through a trust model based on code signing.

**NaCl**. Native Client (NaCl) [3] is a system introduced by Google in 2011 that allows for the execution of machine code on the web. The sandboxing model implemented by NaCl enables the coexistence of NaCl code with sensitive data within the same process. However, NaCl is specifically designed for the x86 architecture, limiting its portability. To address this limitation, Google introduced Portable Native Client (pNaCl) [21] in 2013. pNaCl builds upon NaCl's sandboxing techniques and uses an LLVM bitcode subset as an interchangeable format, allowing for the portability of applications across different architectures. However, pNaCl does not significantly improve compactness and still exposes details specific to compilers and architectures, like the call stack layout. The portability of NaCl and pNaCl is also limited since they are only supported in Google Chrome.

**Asm.js**. Asm.js [4], which was introduced by Mozilla in 2013, is a strict subset of JavaScript that can be used as an efficient compilation target for high-level languages like C and C++. Through the Emscripten toolchain [22], these languages can be compiled to asm.js and subsequently executed on modern JavaScript execution engines, benefitting from sophisticated Just In Time (JIT) compilers. This allows for near-native performance. However, the nature of asm.js as a strict subset of JavaScript means that any extension of its features requires modifications to JavaScript first, followed by ensuring that these changes are compatible with asm.js, which makes the features challenging to implement effectively.

**Java and Flash**. It is also worth noting that Java and Flash were among the first technologies to be used on the web, being released in 1995 and 1996, respectively, [23,24]. They offered managed runtime plugins; however, neither was capable of supporting high-performance, low-level code. Moreover, their usage has declined due to security vulnerabilities and performance issues.

### 2.2. WebAssembly

**Overview**. WebAssembly is a technology that aims to address performance, compatibility, and security issues that have plagued previous approaches. It was developed by a consortium of tech companies, including Mozilla, Microsoft, Apple, and Google, and was released in 2017 [25]. WebAssembly has since gained widespread adoption and is currently supported by 96% of all browser instances [5]. Additionally, it is an official World Wide

Web Consortium (W3C) standard [26], and is natively supported on the web. An overview of WebAssembly is given in Figure 1.

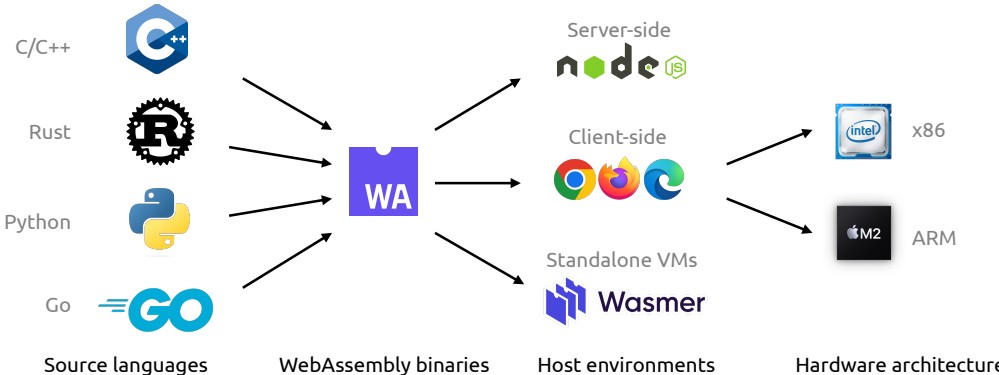

**Figure 1.** WebAssembly serves as the intermediate bytecode bridging the gap between multiple source languages and host environments. The host environments compile the WebAssembly binaries into native code for the specific hardware architecture.

WebAssembly is a low-level bytecode language that runs on a stack-based Virtual Machine (VM). More specifically, instructions push and pop operands to the evaluation stack. This architecture does not use registers; instead, values are stored in global variables that are accessible throughout the entire module or in local variables that are scoped to the current function. The VM manages the evaluation stack, global variables, and local variables.

**Host Environment**. WebAssembly modules run within a host environment, which provides the necessary functionality for the module to perform actions such as I/O or network access. In a browser, the host environment is provided by the JavaScript engine, such as V8 or SpiderMonkey. WebAssembly exports can be wrapped in JavaScript functions using the WebAssembly JavaScript API [27], allowing them to be called from JavaScript code. Similarly, WebAssembly code can import and call JavaScript functions. Other host environments for WebAssembly include server-side environments like Node.js [28] and stand-alone VMs with accompanying APIs. For instance, the WebAssembly System Interface (WASI) [29] allows WebAssembly modules to access the file system.

**Module**. WebAssembly modules serve as the fundamental building blocks for deployment, loading, and compilation. A module contains definitions for types, functions, tables, memories, and globals. In addition, a module can declare imports and exports, as well as provide initialization through data and element segments or a start function.

**Compilation**. Languages like C, C++, and Rust can be compiled into WebAssembly since it is designed as a compilation target. Toolchains like Emscripten [22] or wasm-pack [30] can be used to compile these languages to WebAssembly. The resulting binary is in the wasm binary format, but can also be represented in the equivalent human-readable text format called wat. A module corresponds to one file. The WebAssembly Binary Toolkit (WABT) [31] provides tools for converting between wasm and wat representations, as well as for the de-compilation and validation of WebAssembly binaries.

**Use Cases**. WebAssembly has been adopted for various applications on the web due to its near-native execution performance, such as data compression, game engines, and natural language processing. However, the usage of WebAssembly is not only limited to the web. It is also being extended to desktop applications [6], mobile devices [7], cloud computing [8], IoT [12,13], and embedded devices [14].

2.2.1. Security

**Environment**. WebAssembly modules run in a sandboxed environment which uses fault isolation techniques to separate it from the host environment. As a result of this, modules have to go through APIs to access external resources. For instance, modules

that run in the web browser must use JavaScript APIs to interact with the Document Object Model (DOM). Similarly, stand-alone runtimes must use APIs, like WASI, to access system resources like files. In addition to this, modules must adhere to the security policies implemented by its host environment, such as the Same Origin Policy (SOP) [32] enforced by web browsers, which restricts the flow of information between web pages from different origins.

**Memory**. Unlike native binaries, which have access to the entire memory space allocated to the process, WebAssembly modules only have access to a contiguous region of memory known as linear memory. This memory is untyped and byte-addressable, and its size is determined by the data present in the binary. The size of linear memory is a multiple of a WebAssembly page, each being 64 KiB in size. When a WebAssembly module is instantiated, it uses the appropriate API call to allocate the memory that is needed for its execution. The host environment then creates a managed buffer, typically an `ArrayBuffer`, to store the linear memory. This means that the WebAssembly module accesses the physical memory indirectly through the managed buffer, which ensures that it can only read and write data within a limited area of the memory.

**Control Flow Integrity**. WebAssembly enforces structured control flow, organizing instructions into well-nested blocks within functions. It restricts branches to the end of surrounding blocks or within the current function, with multi-way branches targeting only pre-defined blocks. This prevents unrestricted go-tos or executing data as bytecode, eliminating attacks like shellcode injection or the misuse of indirect jumps. Additionally, execution semantics ensure safety for direct function calls through explicit indexing and protected returns with a call stack. Indirect function calls undergo runtime checks for type signatures, establishing coarse-grained, type-based control-flow integrity. Additionally, the LLVM compiler infrastructure has been adapted to include a fine-grained control flow integrity feature, specifically designed to support WebAssembly [33].

### 2.2.2. Vulnerabilities

Inherent vulnerabilities in the source code can lead to subsequent vulnerabilities in WebAssembly modules [15]. Specifically, buffer overflows in memory-unsafe languages like C and C++ can overwrite constant data or the heap in WebAssembly modules. Despite WebAssembly's sandboxing, these vulnerabilities allow malicious script injection into the module's data section, which is accessible via JavaScript APIs. An example of this is the Emscripten API [22], which allows developers to access data from WebAssembly modules and inject these into the DOM, which can lead to Cross Site Scripting (XSS) attacks [34]. Notably, two-thirds of WebAssembly binaries are compiled from memory-unsafe languages [16], and these attacks have been shown to be practical in real-world scenarios [15]. For instance, Fastly, a cloud platform that offers edge computing services, experienced a 45 min disruption on 8 June 2021, when a WebAssembly binary with a vulnerability was deployed [35]

### 2.2.3. Smart Contracts

Smart contracts are computer programs that are stored on a blockchain, designed to automatically execute once predetermined conditions are met, eliminating the need for intermediaries. As initially proposed by Nick Szabo in 1994 [36], long before the advent of Bitcoin, they have since gained widespread popularity alongside the rise of blockchain technology and cryptocurrencies. The inherent properties of blockchain, such as transparency, security, and immutability, make smart contracts particularly appealing for cryptocurrency transactions. This ensures that once the terms of the contract are agreed upon and coded into the blockchain, they can be executed without the possibility of fraud or third-party interference. Smart contracts can facilitate a variety of transactions, from the transfer of cryptocurrency between parties to the automation of complex processes in finance, real estate, and beyond. Due to its near-native performance, WebAssembly has been adopted by blockchain platforms, such as EOSIO [10] and NEAR [11], as their smart

contract runtime. Ethereum has included WebAssembly in the roadmap for Ethereum 2.0, positioning it as the successor to the Ethereum Virtual Machine (EVM) [9].

However, as with any technology, smart contracts are not without their challenges and vulnerabilities. The immutable nature of blockchain means that once a smart contract is deployed, it cannot be modified, making the correction of vulnerabilities in its code challenging. Several incidents have highlighted the potential financial and security risks associated with vulnerabilities in WebAssembly smart contracts. For instance, random number generation vulnerabilities led to the theft of approximately 170,000 EOS tokens [17]. Similarly, the fake EOS transfer vulnerability in the EOSCast smart contract has led to the theft of approximately 60,000 EOS tokens [18]. The forged transfer notification vulnerability in EOSBet has resulted in the loss of 140,000 EOS tokens [18]. Based on the average price of EOS tokens at the time of the attacks, the combined financial impact of these three vulnerabilities amounted to roughly USD 1.9 million. Additionally, around 25% of WebAssembly smart contracts have been found to be vulnerable [37].

### 2.2.4. Cryptojacking

Cryptojacking, also known as drive-by mining, involves using a website visitor's hardware resources for mining cryptocurrencies without their consent. Previously, cryptojacking was implemented using JavaScript. However, in recent years WebAssembly has been utilized due to its computational efficiency. The year after WebAssembly was released, there was a 459% increase in cryptojacking [38]. The following year, researchers found that over 50% of all sites using WebAssembly were using it for cryptojacking [19]. To counter this trend, researchers developed several static and dynamic detection methods for identifying WebAssembly-based cryptojacking.

While there are theories suggesting that WebAssembly can be used for other malicious purposes, like tech support scams, browser exploits, and script-based keyloggers [39], evidence of such misuse in real-world scenarios has not been documented. As a result, there are no analysis techniques for detecting such malicious WebAssembly binaries. Consequently, discussions about malicious WebAssembly binaries in this paper mainly refer to crypto mining binaries.

## 3. Related Work

This section discusses related work. Specifically, related studies are presented and the differences between those studies and our paper are discussed.

In a similar vein to this paper, Kim et al. [40] survey the various techniques and methods for WebAssembly binary security. However, their focus is on general security techniques for WebAssembly, while our paper specifically focuses on analysis techniques for WebAssembly. We both discuss cryptojacking detection and vulnerability detection for WebAssembly, but we go further by also examining vulnerability analysis for WebAssembly smart contracts. Additionally, we use different classification systems and performance metrics.

Tekiner et al. [41] focus on surveying cryptojacking detection techniques by strictly evaluating and comparing state-of-the-art methods. In contrast, our paper examines analysis techniques for WebAssembly, including cryptojacking detection, vulnerability analysis for WebAssembly binaries, and vulnerability analysis for WebAssembly smart contracts. We also use different classification systems and performance metrics.

Romano et al. [42] investigate the bugs in WebAssembly compilers, specifically examining the Emscripten [22], AssemblyScript [43], and WebAssembly-Bindgen [44] compilers. They discover bugs in the Emscripten compiler that could potentially cause significant security issues. Our work, on the other hand, focuses on security in WebAssembly binaries using analysis techniques, rather than examining the security of the compilers themselves.

## 4. Methodology

This section outlines the methodology used to conduct the systematic review. The literature review aims to identify, evaluate, and synthesize the findings from previous studies on vulnerability analysis, malicious WebAssembly binaries, and smart contracts within the WebAssembly context.

### 4.1. Search Strategy

The primary sources for the literature search were Google Scholar and Scopus. The search terms used were a combination of keywords related to WebAssembly and its security aspects. These included "WebAssembly", "WebAssembly security", "WebAssembly vulnerability analysis", "malicious WebAssembly binaries", "cryptojacking", and "WebAssembly smart contracts", as well as their synonyms and related terms. Boolean operators (AND, OR) were used to refine the search queries, aiming to capture a broad spectrum of relevant research.

The search was actively conducted from August 2022 to December 2022. Given the emerging nature of WebAssembly and its security landscape, we did not apply any publication date restrictions in our search criteria. This approach allowed us to include all relevant studies, from the inception of WebAssembly to the latest advancements, ensuring our review reflects the complete historical and contemporary context of WebAssembly security research.

### 4.2. Selection Process

The selection process was designed to include studies that had developed analysis methods and tools specifically for WebAssembly. Given the novelty of the field, all studies implementing such techniques were considered. However, exclusions were made for papers not directly related to vulnerability analysis, malicious WebAssembly binaries, or smart contracts. Furthermore, only peer-reviewed journal articles were included, ensuring the credibility and reliability of the results.

An additional inclusion criterion was the application of the proposed analysis technique on at least ten samples. This criterion was set to ensure that included studies had their methods tested adequately, providing a measure of reliability and applicability of the findings. The sample size for each method is presented in the following sections in the aim of illustrating the extent to which each technique has been tested and validated.

### 4.3. Data Extraction and Analysis

Data extraction was performed on the selected papers, focusing on implementation details, the application domain (vulnerability analysis, detection of malicious binaries, or smart contracts), sample size, and the performance of the methods. The papers used different metrics for evaluating the performance of their methods, so we converted their results into a standardized set of metrics to have a basis for comparison.

For evaluating the performance of the analysis techniques, we opted to use precision, recall, and $F_1$ scores. Precision measures the proportion of retrieved items that are relevant, while recall measures the proportion of relevant items that are retrieved. A high number of false positives will decrease the precision, while a high number of false negatives will decrease the recall. The $F_1$ score is the harmonic mean of precision and recall and provides a way to combine these two metrics into a single value.

These metrics are mathematically defined as:

$$\text{Precision} = \frac{\text{TP}}{\text{TP} + \text{FP}} \tag{1}$$

$$\text{Recall} = \frac{\text{TP}}{\text{TP} + \text{FN}} \tag{2}$$

$$\text{F}_1 \text{ score} = 2 \times \frac{\text{Precision} \times \text{Recall}}{\text{Precision} + \text{Recall}} \tag{3}$$

These metrics are used instead of accuracy because they are better suited for evaluating the performance of analysis techniques in the presence of imbalanced datasets, which have been common in the literature. In addition to these metrics, the performance of static-based methods has been evaluated using the detection time, while the performance of dynamic-based methods has been evaluated using the runtime overhead. These metrics provide a way to compare the different analysis techniques and assess their relative strengths and weaknesses.

## 5. Analysis Techniques for WebAssembly

This section presents the results of our literature review on analysis techniques for WebAssembly. The techniques can be broadly classified into three categories:

1. Detecting malicious WebAssembly binaries (Section 5.1);
2. Detecting vulnerabilities in WebAssembly binaries (Section 5.2);
3. Detecting vulnerabilities in WebAssembly smart contracts (Section 5.3).

The analysis techniques can be further classified as either static, dynamic, or hybrid methods. Some techniques are static-based, meaning that they analyze the WebAssembly bytecode or intermediate representation without executing the binary. Other techniques are dynamic-based, meaning they analyze the behavior of the WebAssembly binary as it is being executed. Finally, some techniques are hybrid-based, meaning that they combine both static and dynamic analysis to detect potential security issues. The classification is summarized in Table 1.

**Table 1.** Classification of analysis techniques for WebAssembly.

| Category | Static Analysis | Dynamic Analysis | Hybrid Analysis |
|---|---|---|---|
| Detecting malicious WebAssembly binaries (Section 5.1) | MineSweeper [45] | SEISMIC [46] | |
| | MinerRay [47] | RAPID [48] | |
| | MINOS [49] | OutGuard [50] | |
| | | MineThrottle [51] | |
| | | CoinSpy [52] | |
| Detecting vulnerabilities in WebAssembly binaries (Section 5.2) | Wassail [53] | Szanto et al. [54] | WASP$_2$ [55] |
| | Wasmati [56] | TaintAssembly [57] | |
| | WASP$_1$ [58] | Wasabi [59] | |
| | | Fuzzm [60] | |
| | | WAFL [61] | |
| Detecting vulnerabilities in WebAssembly smart contracts (Section 5.3) | EVulHunter [62] | EOSFuzzer [18] | |
| | WANA [63] | WASAI [64] | |
| | EOSAFE [37] | | |
| | EOSIOAnalyzer [65] | | |

*5.1. Detecting Malicious WebAssembly Binaries*

As previously mentioned, WebAssembly has been used by adversaries for malicious purposes, like cryptojacking (Section 2.2.4). To protect against such attacks, several detection techniques have been proposed. In this section, we will review these techniques and evaluate their performance.

Techniques based on static analysis (Section 5.1.1) and dynamic analysis (Section 5.1.2) are discussed in the following sections. Additionally, a comparative analysis of the detection techniques is presented (Section 5.1.3).

5.1.1. Static Analysis

**MineSweeper**. MineSweeper [45] detects cryptojacking based on the presence of cryptographic functions in WebAssembly binaries. MineSweeper implements two variants: The first variant is specialized for detecting the CryptoNight algorithm [66], which is commonly used for cryptomining, while the second variant is more generic and can detect any cryptographic function that may be used for cryptomining. A cryptographic fingerprint is computed by counting the number of cryptographic operations in each function in the WebAssembly binary. In the case of the CryptoNight variant, these fingerprints are then compared with the fingerprints of the primitive components of the CryptoNight algorithm. In the generic case, a candidate function is labeled as a cryptographic function if the amount of cryptographic operations exceeds a threshold. The authors conducted experiments to validate the effectiveness of their method, achieving 100% recall and precision for both variants. However, it is worth noting that the authors theorized a potential limitation of the method, suggesting that, under certain conditions, it might produce false positives. This is because benign programs, such as games and cryptographic libraries, also utilize cryptographic functions, which could theoretically be misidentified by MineSweeper as indicators of cryptojacking.

**MinerRay**. MinerRay [47] constructs and analyzes an Inter-Procedural Control Flow Graph to detect cryptojacking. MinerRay first converts JavaScript and asm.js code into WebAssembly binaries. Then, the WebAssembly binaries are translated into an intermediate representation, from which Intra-Procedural Control Flow Graphs are constructed for each function. These Intra-Procedural Control Flow Graphs are then linked together to create an Inter-Procedural Control Flow Graph that represents the entire program. MinerRay uses the Inter-Procedural Control Flow Graph to identify potential hashing algorithms by analyzing the control flow of the program and looking for patterns that match the semantics of hashing functions. To determine whether the user is informed about cryptomining, MinerRay employs a dynamic approach that explores the `onclick` events of HTML objects, which may instantiate WebAssembly cryptominers. It then checks whether the WebAssembly APIs, such as `WebAssembly.instantiate`, can be invoked. Out of 901 websites with cryptominers, the authors found that only 16 websites informed users of the background crypto mining and just three of those asked for consent before starting the mining process.

**MINOS**. MINOS [49] uses an image-based classification deep learning approach to identify cryptojacking. First, MINOS converts the WebAssembly binary into a grayscale image. This image is then used as input to a Convolutional Neural Network (CNN), which has been trained on a comprehensive dataset of malicious and benign WebAssembly binaries. The CNN attempts to determine whether the WebAssembly binary performs cryptojacking based on the patterns it observes in the grayscale image. An advantage of MINOS is that it is lightweight and can detect cryptojacking in under a second. This makes it a useful tool for real-time cryptojacking detection.

5.1.2. Dynamic Analysis

**SEISMIC**. SEISMIC [46] uses signature-matching to identify cryptojacking. It adopts an In-Line Reference Monitor (IRM) approach, which involves dynamically computing the semantic features of the WebAssembly binary at runtime. To this end, an instruction counter is inserted into the global section of the WebAssembly binary for each instruction

to be profiled. The semantic features of the WebAssembly binary are then computed using the aforementioned instruction counters. To identify cryptojacking, the computed semantic features of the WebAssembly binary are compared with the semantic signatures of known mining binaries. This approach was found to be accurate in detecting cryptojacking, but it imposes a significant runtime overhead, which can affect the performance of the WebAssembly application.

**RAPID**. RAPID [48] identifies cryptojacking by monitoring JavaScript API calls and system resource usage. To this end, JavaScript API usage is collected using Chrome debugging features. The system resources, that is, the memory, network, and processor usage are collected by executing a Chromium instance inside a docker container and collecting the data through the docker stats API [67]. Then, a Support Vector Machine (SVM) is employed as a classification model.

**OutGuard**. OutGuard [50] uses features related to the JavaScript runtime execution, event loads, networking, and cryptojacking libraries to detect cryptojacking. Specifically, the number of web workers and parallel tasks, the existence of WebAssembly modules, WebSockets and hashing algorithms, and the usage of PostMessage- and MessageLoop event loads are used as the feature set. These seven distinct features are used to build an SVM classification model. A limitation of this approach is that the identification of hashing algorithms is static and does not account for string obfuscation.

**MineThrottle**. MineThrottle [51] uses the frequency distribution of instructions to detect cryptojacking. The idea is that miners execute certain instructions more frequently than benign applications, and this can be used to identify the mining activity. To implement this, MineThrottle first detects potential mining-related code blocks using block-level statistical features and then instruments each block using block-level program profiling. The effective mining speed (i.e., the instructions per cycle) of the WebAssembly program is then periodically calculated, and if it is similar to known mining programs, the program is labeled as a miner.

**CoinSpy**. CoinSpy [52] is a method for detecting cryptojacking by monitoring compute, memory, and network usage from within the browser. The computational behavior is monitored using the JavaScript stack profiler, and memory usage is measured by monitoring the JavaScript heap and WebWorker threads. Network usage is tracked by summing the bytes from all in-flight requests at each millisecond. The key observation used for cryptojacking detection is that compute and memory usage increase significantly when the Proof of Work (PoW) algorithm is executing, and that network usage only increases when the processor is in an idle state. Using these features, a CNN classification model was constructed. The authors argue that CoinSpy should be able to detect future cryptomining algorithms that other dynamic detectors will miss due to their specificity.

### 5.1.3. Comparative Analysis

This section presents the comparative analysis of the detection techniques outlined in the above sections. The results of the analysis are summarized in Tables 2 and 3.

**Table 2.** Data for static detection techniques for identifying malicious WebAssembly binaries.

| Scheme | Feature(s) | Classifier | Dataset | | Performance | | | |
| --- | --- | --- | --- | --- | --- | --- | --- | --- |
| | | | Source | Samples | Precision | Recall | $F_1$ | DT* |
| MineSweeper [45] (2018) | WebAssembly code | Matching or threshold | Alexa 1M | 748 | 100% | 100% | 100% | - |
| MinerRay [47] (2020) | WebAssembly code | ICFG | Alexa 1.2M | 3825 | 99% | 100% | 99% | 1.9 s |
| MINOS [49] (2021) | WebAssembly code | CNN | Tranco 100K, PublicWWW | 682 | 93% | 97% | 95% | 0.0259 s |

* Abbreviations: detection time (DT).

**Table 3.** Data for dynamic detection techniques for identifying malicious WebAssembly binaries.

| Scheme | Feature(s) | Classifier | Dataset | | Performance | | | |
| | | | Source | Samples | Precision | Recall | $F_1$ | Overhead |
|---|---|---|---|---|---|---|---|---|
| SEISMIC [46] (2018) | WebAssembly code, instruction count obtained at runtime | Matching | Asteroids, A-Star, Tanks, Bullet (1000), CoinHive_v0, CoinHive, Basic4GL, HushMiner, CreaturePack FunkyKarts, NFWebMiner, YAZECMiner | 12 | 96% | 100% | 98% | 100% |
| RAPID [48] (2018) | JavaScript API calls, memory, processor and network usage | SVM | Alexa 330K | 71,450 | 97% | 96% | 96% | 9–40% |
| OutGuard [50] (2019) | Parallel tasks, WebAssembly, hashing algorithms, WebSockets, PostMessage event load, MessageLoop event load | SVM, or RF | Alexa 1M, Alexa 600K | 29,700 | 99% | 97% | 98% | 2% |
| CoinSpy [52] (2020) | JavaScript stack execution time, JavaScript heap, network usage | CNN | Alexa 1M, Alexa 100K, PublicWWW | 2000 | Accuracy: 97% | | | 0% |
| MineThrottle [51] (2020) | WebAssembly code | | | | | | | |
| processor usage | Matching | Alexa 1M | 659 | 100% | 98% | 99% | 0% | |

**Dataset**. Most detection techniques have been evaluated using websites collected from the wild, with the Alexa sites being the most commonly used. Only SEISMIC evaluated their method using a curated list of binaries. Most schemes used a sufficient number of samples, but there were some exceptions, such as SEISMIC, MineThrottle, and MINOS, which had a smaller number of samples, potentially affecting the validity of their results.

**Performance**. The performance of the detection techniques was evaluated using metrics such as precision, recall, $F_1$ score, overhead, and detection time. Among the static-based methods, MineSweeper and MinerRay had the highest $F1_1$ scores, while MINOS had the lowest. However, MINOS also had the fastest detection time, making it suited for real-time cryptojacking detection. MineThrottle, Outguard, and SEISMIC had the highest $F_1$ scores among the dynamic-based methods. However, SEISMIC also had the highest overhead. In contrast, MineThrottle and Outguard had negligible overhead.

## 5.2. Detecting Vulnerabilities in WebAssembly Binaries

Although WebAssembly was designed with security in mind, vulnerabilities still exist (Section 2.2.2). As a result, various techniques for detecting vulnerabilities in WebAssembly binaries have been proposed. This section presents these techniques and discusses their versatility, which is determined by factors such as compatibility with different runtimes, support for the WASI, and whether they require high-level source code for analysis.

Techniques based on static analysis (Section 5.2.1), dynamic analysis (Section 5.2.2), and hybrid analysis (Section 5.2.3) are discussed in the following sections. Additionally, a comparative analysis of the detection techniques is presented (Section 5.2.4).

### 5.2.1. Static Analysis

**Wassail**. Wassail [53] was the first static analysis method for detecting vulnerabilities in WebAssembly binaries. It uses a compositional, summary-based analysis approach that strictly focuses on information flow. For each WebAssembly function, it computes a summary that describes how information flows within that function, and these summaries are then used during the subsequent analysis of function calls. The information flow analysis is expressed as a data flow analysis on a Control Flow Graph (CFG), and the information flow of the entire program is approximated by composing the function summaries. The authors claim that similar approaches have been shown to scale well [68,69], but the scalability of Wassail has not been evaluated.

**Wasmati**. Wasmati [56] detects vulnerabilities in WebAssembly binaries by constructing a Code Property Graph (CPG). Vulnerabilities are detected by searching for specific patterns in the CPG sub-graphs, which include the execution order, execution path, data dependencies, and control flows. An issue with this approach is that the number of nodes in the CPG grows rapidly because the target of indirect calls cannot be determined statically. To address this, Wasmati optimizes the CPG generation process by adding additional annotations, caching intermediate results, and using efficient graph traversal algorithms. As a result, the authors found that constructing the CPG only took an average of 58 s per binary. They also found that Wasmati was able to effectively find vulnerabilities in WebAssembly binaries while providing a low false positive rate.

**WASP$_1$**. WASP$_1$ [58] is a concolic execution engine for WebAssembly modules that can be used for uncovering vulnerabilities and bugs. Concolic execution, which combines concrete execution with symbolic execution and explores one execution path at a time, is employed to explore all the feasible paths of the program. Specifically, symbolic execution is used to generate the concrete inputs for exploring multiple execution paths, to maximize code coverage. To demonstrate the feasibility of uncovering vulnerabilities, the authors constructed WASP-C, a symbolic execution framework for testing C programs using WASP$_1$. WASP-C takes a C program as input, annotates it, compiles it to WebAssembly, and analyzes it using WASP$_1$. They found that WASP-C was effective at uncovering bugs and vulnerabilities. However, a limitation of WASP-C is that it requires a high-level source code to uncover vulnerabilities, meaning it can only be used to analyze open source programs.

### 5.2.2. Dynamic Analysis

**Szanto et al**. Szanto et al. [54] proposed a taint-tracking technique for detecting vulnerabilities in WebAssembly binaries. They developed a VM that runs in native JavaScript and implemented a taint tracking system that allows the user to monitor the flow of sensitive data through the execution of the WebAssembly binary. To this end, they allocate a tainted label for each allocable byte in the memory section and each variable on the stack. This method allows for taint tracking without modifying the structure of the WebAssembly binary. The authors found that the runtime overhead of this method scales mostly linearly, with an overhead of up to 100%.

**TaintAssembly**. TaintAssembly [57] is another technique that uses taint-tracking to detect vulnerabilities in WebAssembly binaries. Unlike Szanto et al., who developed their own VM, TaintAssembly implemented taint-tracking by modifying the V8 JavaScript engine used in Google Chrome and Node.js [28]. TaintAssembly implements basic taint-tracking functionality for variables of type `i32`, `i64`, `f32`, `f64`, as well as tainting in linear memory and a probabilistic variant of taint. However, unlike Szanto et al.'s approach, the structure of the WebAssembly module must be modified before taint labels can be set for all variables. TaintAssembly was able to achieve a runtime overhead of only 5–12%, which is far less than Szanto et al.'s approach.

**Wasabi**. Wasabi [59] is a general-purpose framework for dynamically analyzing Web-Assembly binaries. To this end, Wasabi performs binary instrumentation. Specifically, it inserts calls to analyze functions written in JavaScript into the WebAssembly binary. Then, instruction counting, call graph extraction, memory access tracing, and taint analysis can be performed at runtime. Wasabi also allows for selective instructions; that is, it only instruments instructions that are relevant for a particular analysis. The authors found the runtime overhead to vary between 2% and 163%, depending on the application and instructions being analyzed.

**Fuzzm**. Fuzzm [60] is a binary-only fuzzer for WebAssembly that uses the popular AFL [70] framework. Native AFL compiles applications from source code and inserts code to track path coverage. However, since Fuzzm is a binary-only fuzzer, it does not have access to the source code. To provide coverage information for the AFL fuzzer, Fuzzm uses static binary instrumentation to insert code at all branches, generating AFL-compatible coverage information. The authors found that Fuzzm is effective and imposes a low runtime overhead. Additionally, its implementation is not tied to a specific runtime. Fuzzm also implements a canary-based protection mechanism to prevent memory corruption vulnerabilities.

**WAFL**. WAFL [61] is also a binary-only fuzzer for WebAssembly. It uses the AFL++ [71] framework, a community-driven fork of AFL. To generate coverage for the AFL++ fuzzer, they implement a set of patches to the WAVM [72] runtime. The WAVM runtime uses Ahead-of-time (AOT) compilation, and WAFL also adds lightweight VM snapshots. This makes WAFL performant, in some cases, even outperforming the native AFL x86-64 harnesses compiled from source. However, WAFL is inherently tied to the WAVM runtime, which limits its potential use cases.

### 5.2.3. Hybrid Analysis

**WASP$_2$**. WASP$_2$ [55] detects vulnerabilities in WebAssembly binaries based on known vulnerabilities. It does this by analyzing the static and dynamic features of the WebAssembly binary and comparing this with known vulnerabilities. Specifically, WASP$_2$ trains a deep learning vulnerability classification model by mapping the static features of known vulnerable binaries in x86 or ARM to static features in the corresponding WebAssembly binary representation. Then, the model is used to statically analyze the WebAssembly binary. Finally, the identified vulnerable subroutines are dynamically analyzed using Wasabi [59]. The authors found that WASP$_2$ is able to accurately find known vulnerabilities in WebAssembly binaries.

### 5.2.4. Comparative Analysis

This section presents the comparative analysis of the detection techniques outlined in the above sections. The results from the analysis are summarized in Table 4.

**Runtime Compatibility**. The versatility of the proposed detection techniques is determined by their runtime compatibility, WASI support, and whether they require a high-level source code for their analysis. Some schemes, like TaintAssembly, are inherently tied to a specific runtime, which limits their usefulness. Other schemes, like Wasabi, are not tied to any specific runtime and can be applied more generally. Additionally, schemes that do not support WASI, like Szanto et al.'s method, are fundamentally limited since they cannot analyze most WebAssembly binaries used on servers or embedded devices. Finally, schemes that require high-level source code for analysis, like WASP$_1$, are limited to the analysis of open source projects.

**Overhead**. The overhead for dynamic detection methods varies greatly. Wasabi and Szanto et al. have the highest overhead, reaching up to 163% and 100%, respectively. Szanto et al.'s method has a constant overhead, while Wasabi's overhead varies depending on the type and number of instructions being analyzed. This means that Wasabi's overhead can be as low as 2% in practice. In contrast, TaintAssembly and Fuzzm demonstrate much lower overheads, within the range of 5-12% and 5-6%, respectively. The lower overhead associated

**Table 4.** Data for detecting vulnerabilities in WebAssembly binaries.

| Type | Scheme | Technique | Runtime | Binary-Only | WASI Support | Overhead |
|------|--------|-----------|---------|-------------|--------------|----------|
| Static | Wassail [53] (2020) | Compositional information flow | - | ✓ | - | - |
| | Wasmati [56] (2022) | CPG | - | ✓ | - | - |
| | WASP$_1$ [58] (2022) | Concolic execution | - | ✗ | - | - |
| Dynamic | Szanto et al. [54] (2018) | Taint tracking | WebAssembly VM (custom) | ✓ | ✗ | 100% |
| | TaintAssembly [57] (2018) | Taint tracking | V8 engine (modified) | ✓ | ✓ | 5–12% |
| | Wasabi [59] (2019) | Binary instrumentation | Any runtime | ✓ | ✓ | 2–163% |
| | Fuzzm [60] (2021) | Fuzzing | Any runtime w/ WASI-support | ✓ | ✓ | 5–6% |
| | WAFL [61] (2021) | Fuzzing | WAVM (modified) | ✓ | ✓ | - |
| Hybrid | WASP$_2$ [55] (2021) | Known vulnerabilities | - | ✓ | ✓ | - |

✓: Feature is supported. ✗: Feature is not supported.

with TaintAssembly indicates that integrating taint-tracking into VMs can be a viable option for increased security with low-to-moderate overhead. However, this integration must carefully consider the balance between security improvements and potential performance trade-offs for each specific use case.

### 5.3. Detecting Vulnerabilities in WebAssembly Smart Contracts

Several vulnerabilities have been discovered in WebAssembly smart contracts, leading to significant financial loss (Section 2.2.2). As a result, several techniques for detecting such vulnerabilities have been developed. This section presents these techniques, their capabilities, and their performance.

Techniques based on static analysis (Section 5.3.1) and dynamic analysis (Section 5.3.2) are discussed in the following sections. Additionally, a comparative analysis of the detection techniques is presented (Section 5.3.3).

#### 5.3.1. Static Analysis

**EVulHunter**. EVulHunter [62] was the first static analysis tool designed to detect vulnerabilities in EOSIO smart contracts. It uses the open source analysis framework Octopus [73] to construct a CFG of the smart contract. The CFG is then traversed to detect vulnerabilities based on predefined patterns. Although EVulHunter is effective at detecting fake notification vulnerabilities, it has low precision when detecting fake EOS transfers. The authors believe this is due to the limitations of using predefined patterns and suggest that more advanced analysis techniques, such as symbolic execution, are necessary.

**WANA**. WANA [63] uses symbolic execution and a set of test oracles to detect vulnerabilities in smart contracts. It is cross-platform, meaning that it can detect vulnerabilities in both EOSIO and Ethereum smart contracts. First, Ethereum smart contracts, which are written in Solidity, are converted into Ewasm (Ethereum-flavored WebAssembly) using

the SOLL [74] compiler. Then, the symbolic execution engine traverses the paths of the WebAssembly binary. WANA performs vulnerability analysis based on the data collected during symbolic execution using the proposed test oracles. Unlike EVulHunter, WANA can also detect blockinfo dependency vulnerabilities and effectively detect fake EOS transfer vulnerabilities.

**EOSAFE**. EOSAFE [37] also uses symbolic execution to detect vulnerabilities in EOSIO smart contracts. Unlike WANA, EOSAFE addresses the problem of path explosion, where the number of feasible paths in a program grows exponentially with the program size. To mitigate this issue, EOSAFE allows users to set the call depth and timeout parameters that are used when symbolically executing the program. Additionally, during vulnerability detection, EOSAFE first identifies valuable functions (i.e., functions that have the ability to invoke actions or change on-chain state) and only analyzes those. This approach allows it to accurately detect more vulnerabilities than previous methods.

**EOSIOAnalyzer**. EOSIOAnalyzer [65] detects vulnerabilities in EOSIO smart contracts by analyzing the ICFG of the program. The ICFG is the combination of the CFG and the Call Graph (CG) of the program, allowing EOSIOAnalyzer to analyze the data propagation relationships between functions when they call each other. After the ICFG is constructed, the WebAssembly code is translated into a high-level intermediate representation. Then, EOSIOAnalyzer applies a data flow analysis algorithm to determine the data propagation relationships between functions. Finally, it identifies suspicious functions and further analyzes their complete execution paths. To address the issue of path explosion, EOSIOAnalyzer implements a call depth threshold.

### 5.3.2. Dynamic Analysis

**EOSFuzzer**. EOSFuzzer [18] uses black-box fuzzing to detect vulnerabilities in EOSIO smart contracts. To this end, EOSFuzzer first performs static analysis on the WebAssembly code and Application Binary Interface (ABI). The results of this analysis are then used to generate fuzzing inputs, which are applied to the smart contract through the Cleos [75] command-line client. Finally, EOSFuzzer performs vulnerability analysis based on test oracles. Although EOSFuzzer was found to be relatively efficient, it has the lowest precision and recall of all methods proposed. Additionally, EOSFuzzer uses random seeds for fuzzing, resulting in low code coverage.

**WASAI**. WASAI [64] uses concolic fuzzing to detect vulnerabilities in EOSIO smart contracts. To address the weaknesses of EOSFuzzer, it strategically generates seeds to aid the fuzzing in exploring as many feasible paths as possible. This is achieved by performing symbolic execution to feedback on the seed mutation. This results in double the code coverage than EOSFuzzer. WASAI was able to detect the most vulnerabilities out of all the proposed techniques. It additionally has high precision and recall and was found to be resilient to code obfuscation.

### 5.3.3. Comparative Analysis

This section presents the comparative analysis of the detection techniques outlined in the above sections. The results from the analysis are summarized in Table 5.

**Vulnerability Detection**. The proposed detection techniques are able to detect different types of vulnerabilities. EVulHunter is only able to detect two out of five types of vulnerabilities, while WANA, EOSIOAnalyzer, and EOSFuzzer are able to detect three out of five types. EOSAFE is able to detect four out of five vulnerabilities. WASAI is the only method that is able to detect all five vulnerabilities, making it the most effective detection method.

**Performance**. WANA and WASAI have the highest $F_1$-score among the proposed detection techniques, with 100% and 99%, respectively. EOSFuzzer and EVulHunter have the lowest $F_1$ scores, with 88% and 93%, respectively. In terms of detection time, EOSIOAnalyzer has the slowest detection time at 7.6 s. In contrast, WANA has a detection time of only 0.21 s while providing high precision and recall.

**Table 5.** Data for detecting vulnerabilities in WebAssembly smart contracts.

| Type | Scheme | Technique | Vulnerability Detection | | | | | Performance | | | |
|------|--------|-----------|------|------|------|------|-------|-----------|--------|-------|------|
| | | | FE* | FN* | BD* | RB* | MAV* | Precision | Recall | $F_1$ | DT* |
| Static | EVulHunter [62] (2019) | CFG | ✓ | ✓ | ✗ | ✗ | ✗ | 89% | 100% | 93% | 1–3 s |
| | WANA [63] (2020) | Symbolic execution | ✓ | ✓ | ✓ | ✗ | ✗ | 100% | 100% | 100% | 0.21 s |
| | EOSAFE [37] (2021) | Symbolic execution | ✓ | ✓ | ✗ | ✓ | ✓ | 100% | 96% | 98% | - |
| | EOSIOAnalyzer [65] (2022) | ICFG | ✓ | ✓ | ✓ | ✗ | ✗ | 93% | 100% | 96% | 7.6 s |
| Dynamic | EOSFuzzer [18] (2020) | Fuzzing | ✓ | ✓ | ✓ | ✗ | ✗ | 88% | 88% | 88% | - |
| | WASAI [64] (2022) | Concolic fuzzing | ✓ | ✓ | ✓ | ✓ | ✓ | 100% | 98% | 99% | - |

\* Abbreviations: fake EOS (FE), fake notification (FN), Blockinfo dependency (BD), rollback (RB), missing authorization verification (MAV), and detection time (DT).

✓: Vulnerability is patched. ✗: Vulnerability is not patched.

## 6. Discussion

This section presents the results of the literature review. It begins by summarizing the key findings, followed by a discussion of the limitations of current analysis techniques, and lastly a discussion on the applicability of the analysis techniques in other domains.

### 6.1. Key Findings

Methods based on static analysis use techniques such as signature matching and symbolic execution which do not require the program to be executed. This allows them to impose a low overhead but can also result in a lower accuracy. For example, MINOS has the fastest detection time at under one second but also has the lowest $F_1$-score of all cryptojacking detection methods. Methods that rely solely on signature or keyword matching can be easily bypassed through the use of obfuscation techniques [46,76]. Additionally, methods that rely on semantic execution may be limited by the path explosion problem.

Methods based on dynamic analysis execute the program in a controlled environment to extract behavioral features that can be used for further analysis. To do this, various techniques such as taint tracking, binary instrumentation, fuzzing, and monitoring system resources have been proposed. This typically results in higher overhead but also a better performance than static-based methods. For example, Wasabi is able to perform heavy-weight dynamic analysis through binary instrumentation, but it might also incur a high overhead. Since dynamic analysis is based on behavioral features, it is less susceptible to evasion through obfuscation techniques. However, it can still be bypassed in some cases, such as by throttling processor usage.

Static and dynamic techniques are not mutually exclusive but complementary techniques. A hybrid approach can leverage the low overhead of static techniques and the high accuracy of dynamic techniques. In such an approach, static techniques can be used to identify candidate functions and dynamic techniques can then be employed to analyze them accurately. Currently, only $WASP_2$ employs such a hybrid approach.

### 6.2. Limitations

The evaluation strategies for each detection method differ substantially. Some methods are evaluated using imbalanced datasets, while others use balanced datasets. Additionally, the sample sizes used for evaluation also differ between the detection methods. To address these variations, we used precision, recall, and $F_1$ as performance metrics instead of

accuracy. However, variations in evaluation strategies can still impact the validity of the results. For example, EVulHunter reported an $F_1$-score of 93%, but other studies found its $F_1$ score to be 17% and 23% [18,65]. However, using the results from these complementary studies may further threaten the validity of the results as they may contain implementation bugs.

Many cryptojacking detection methods do not distinguish between cryptomining and cryptojacking. Cryptomining refers to the use of a user's resources to mine cryptocurrency with the user's explicit consent. This has been used as an alternative revenue source by organizations such as UNICEF [77]. Cryptojacking, on the other hand, refers to the use of a user's resources to mine cryptocurrency without their explicit consent. Currently, only MinerRay is able to differentiate between these two activities. This differentiation may result in a 1–2% increase in false positives [47].

The datasets used for evaluation can impact the results of the evaluation. Cryptojacking websites often modify or move their scripts to different domains to avoid being blacklisted. Additionally, the CoinHive shutdown [16,78] resulted in a decrease in cryptojacking activity. As a result, the accuracy of cryptojacking detection methods may vary depending on when the dataset was collected.

### 6.3. Applicability of WebAssembly Analysis Techniques in Other Domains

The comparison between WebAssembly and native programs raises important questions about the transferability of analysis techniques across these domains. Specifically, can tools like MineSweeper, designed for WebAssembly, be adapted to detect malicious application software in native binaries? Conversely, can traditional virus scanning technologies be effectively used for scanning malicious WebAssembly binaries?

Tools like MineSweeper focus on analyzing WebAssembly binaries for patterns indicative of cryptojacking activities, leveraging the specific structure and execution model of WebAssembly. Adapting such tools for native programs would require addressing the broader spectrum of malicious behaviors that native applications might exhibit, along with the consideration of different binary formats, execution flows, and interaction with the operating system. Thus, analysis techniques for WebAssembly binaries are likely not directly transferable to native programs.

Traditional virus scanning technologies are designed to detect a wide range of malicious signatures and behaviors in native applications. These technologies could potentially identify known malicious patterns or signatures within WebAssembly binaries. However, the effectiveness of this approach may be limited by the specific WebAssembly architecture. That is, the specific exploitation techniques and vulnerabilities relevant to WebAssembly might not align with those typically encountered in native applications, and vice versa.

### 7. Conclusions

In this paper, we conducted a comprehensive review of analysis techniques for WebAssembly. To this end, we constructed a taxonomical classification and applied it to analysis techniques proposed in the literature. We classified the techniques into three categories: detecting malicious WebAssembly binaries, detecting vulnerabilities in WebAssembly binaries, and detecting vulnerabilities in WebAssembly smart contracts. We analyzed these techniques using quantitative data and discussed their strengths and weaknesses. Then, key findings and limitations were presented. Specifically, we found that static methods have low overhead but lower accuracy, while dynamic analysis has higher overhead but higher accuracy. We also identified potential areas for future research, including the security of WebAssembly in non-web environments, analysis techniques for malicious WebAssembly binaries, the feasibility of obfuscating WebAssembly code, and the prevalence of WebAssembly-based cryptojacking on the web. This paper provides a valuable contribution to the field by offering a comprehensive understanding of current analysis techniques for WebAssembly, including their use cases and limitations, as well as suggestions for future research.

### Research Directions

Generally, the proposed WebAssembly analysis techniques are focused on the web environment. As WebAssembly is being extended for use beyond the web, current analysis techniques do not cover all possible use cases. There have been studies on the use of WebAssembly in non-web environments [79], but few have specifically focused on its security in these contexts. Further research addressing the security of WebAssembly in non-web environments is needed.

The proposed detection techniques for detecting malicious WebAssembly binaries are biased towards cryptojacking. WebAssembly can also be used for other malicious purposes, like tech support scams, browser exploits, and script-based keyloggers [39]. Currently, there are no methods for detecting these types of malicious uses of WebAssembly. Further research is encouraged in this direction.

The proposed methods for detecting cryptojacking can be circumvented through code obfuscation, which has previously rendered static detection methods obsolete [80]. The obfuscation of WebAssembly code is common on the web [16,45]. However, only one preliminary study [76] has investigated the feasibility of obfuscation for WebAssembly, and the researchers only evaluated it using one static detection technique. The effects on dynamic detection techniques were not explored. Additionally, the study used a small dataset, potentially undermining the validity of the results. Some authors of cryptojacking detection techniques argue that obfuscation is impractical due to the added runtime overhead and the resulting decrease in revenue from reduced hash rates. However, the effects of the obfuscated WebAssembly code on runtime and hash rates have not been studied. More research in this area is needed.

The prevalence of WebAssembly-based cryptojacking on the web is unclear. There have been two studies on this topic, one by Musch et al. [19] in 2018 and the other by Hilbig et al. [16] in 2021. Musch et al. found that over 50% of sites using WebAssembly were doing so for cryptojacking, while Hilbig et al. found that this number had been marginalized to 1%. This decrease was attributed to the shutdown of CoinHive, which is supported by other studies [78]. However, even after the shutdown of CoinHive, other studies have found the prevalence of WebAssembly-based cryptojacking to be as high as 10% [49]. Moreover, Hilbig et al. used VirusTotal [81] for detecting cryptojacking, which has been proven to be easily bypassed through code obfuscation [46]. Therefore, the results of this study may be inaccurate due to false negatives. Further research in this area is needed.

**Author Contributions:** Conceptualization, H.H.; methodology, H.H.; investigation, H.H.; methodology, H.H.; data curation, H.H.; visualization, H.H.; writing—original draft preparation, H.H.; writing—review and editing, H.H. and D.M.; supervision, D.M. All authors have read and agreed to the published version of the manuscript.

**Funding:** This research received no external funding

**Data Availability Statement:** Not applicable.

**Conflicts of Interest:** The authors declare no conflicts of interest.

**Abbreviations**

The following abbreviations are used in this manuscript:

| | |
|---|---|
| ABI | Application Binary Interface |
| AOT | Ahead-of-Time |
| CFG | Control Flow Graph |
| CG | Call Graph |
| CNN | Convolutional Neural Network |
| COM | Component Object Model |
| CPG | Code Property Graph |
| DOM | Document Object Model |
| EVM | Ethereum Virtual Machine |
| ICFG | Inter-Procedural Control Flow Graph |
| IRM | In-Line Reference Monitor |
| JIT | Just in Time |
| NaCl | Native Client |
| pNaCl | Portable Native Client |
| PoW | Proof of Work |
| SOP | Same Origin Policy |
| SVM | Support Vector Machine |
| VM | Virtual Machine |
| W3C | World Wide Web Consortium |
| WABT | WebAssembly Binary Toolkit |
| WASI | WebAssembly System Interface |
| XSS | Cross-Site Scripting |

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
