# Peer review of "SoK: Analysis Techniques for WebAssembly"

_futureinternet, doi:10.3390/fi16030084_

Round 1
Reviewer 1 Report
Comments and Suggestions for Authors
This article conducts a literature review and classification evaluation of WebAssembly. The reviewer provides the following questions and comments:
(1) The content designed in this article is concentrated and rich. However, what is the significance of the discussion collected by the author for sections such as vulnerability detection?
(2) The relationship between the host environment in Figure 1 is more complex. If this differs from this article's focus, the reviewer recommends that the author redesign the diagram and try to bypass the server and browser.
(3) Source language, front-end and back-end of the Web, cloud computing, and edge computing are all related to the focus of this article. The author should more fully enumerate the contribution of this paper (i.e., collected problems and possible solutions).
(4) The author should indicate future work and research directions at the end of the conclusion chapter.
That is all, thanks.
Reviewer 2 Report
Comments and Suggestions for Authors
The authors present an overview of different tools for analyzing WebAssembly programs with respect to detect (potentially illegitimate) cryptomining, vulnerabilities from memory-unsafe languages and violations of smart contracts. The study results seem interesting, and the text is well-structured and easily readable.
My main concern is, however, that the methods are not described in sufficient detail. The paper presents a systematic review but it is not possible to judge how systematic this review was. The paper must be extended by a methods section that describes how the systematic review was conducted and how relevant tools were collected.
The paper should explain what smart contracts are. A basic understanding of basic contracts is necessary to understand the paper.
The should explain the mechanisms how attacks on vulnerable WebAssembly programs can work. Explain why they are susceptible to attacks from non-memory-safe languages only, and what can be the result of such an attack. After all, the paper explains that the WebAssemblys are sandboxed. Explain how this sandboxing is lacking.
In the early parts of the paper, several sentences are – for my taste – repeated a bit too much. But I accept that this might be a matter of taste. I just wanted to point it out.
I would like to see a short discussion of whether WebAssembly and native programs are comparable in such a way that results can be transferred between the two domains. For example, could MineSweeper also be used to detect malicious application software? Vice versa: Would it make sense to employ virus scanning technology to detect malicious WebAsssembly programs? Of course, such discussion might come to the conclusion that such approach is limited, and that, in fact, better runtime sandboxes are needed. Or may be not? I would just like to see this discussed.
I would also like to see a discussion of whether taint-tracking should be included in VMs (and sandboxes) by default.
I also ask myself: If a malicious activity is detected only a runtime, and the sandbox does not immediately and automatically stop the execution, is it not too late then?
The text creates the impressions that there is only one kind of malicious WebAssemblies, namely cryptojacking. I understand that indeed cryptojacking is very important. But is this the only kind of malicious software that is addressed?
Table 3 and Table 4 have the same caption or combine in one table. Obviously, caption of Table 4 is wrong.
It is unclear how the performance of the presented tools was evaluated (e.g., Table 2 + Table 3). Or are the values just reported from the original publications? Please describe the method.
Round 2
Reviewer 1 Report
Comments and Suggestions for Authors
(1) The author should include the main results and findings in the Abstract (preferably numerical values).
(2) At the end of the first chapter, the author should preview the contents of the remaining chapters.
(3) The related work in Chapter 3 should be compared with the results of this study (preferably in the form of a table).
That's all. Thanks.
Reviewer 2 Report
Comments and Suggestions for Authors
This is my second review of the submission. Thank you for the modifications, and I think that the paper can be accepted with these amendments.
Regarding the new methodology section, the paper now states: “These included […]. Boolean operators (AND, OR) were used to refine the search queries, aiming to capture a broad spectrum of relevant research. The search was conducted without a specific time frame to include all available literature due to the novelty of the field.” => All three sentences leave a lot of room for speculation: included keywords (but there may be more?), boolean operators were applied (but how and to form what concrete queries?), no time frame (not known anymore? Until today? Please provide at least an end date, when it was done.). How many results were there? Were results pre-filtered for irrelevance? Were there iterations? If you can be more precise, be more precise about this process. However, I almost fear that you are unable to reconstruct what you actually did. Therefore, I have to trade off this drawback against the benefits of the paper, and I come to the conclusion that the benefits of the paper in its entirety outweigh these problems with the methodology. However, I ask you to once again revisit what you wrote and be as precise and honest as possible.
Thank you for adding the paragraph on smart contracts. I recommend to start with a more simple introductory sentence about what smart contracts are: “A smart contract is a computer program or a transaction protocol that is intended to automatically execute, control or document events and actions according to the terms of a contract or an agreement. […] Smart contracts are commonly associated with cryptocurrencies and are generally considered a fundamental building block for decentralized finance (DeFi) and NFT applications.” (I copied this from Wikipedia as an example; so don’t directly use this.)
Line 345: “The authors conducted experiments to validate the effectiveness of their method, achieving 100% recall and precision for both variants. However, they acknowledge a potential limitation of MineSweeper: It can produce false positives, as benign programs such as games and cryptographic libraries also use cryptographic functions.” => How can precision be 100%, if there can be false positives? Please rephrase.
Reference 37: is missing a URL: Szabo, N. Smart Contracts, 1994. [Online: Accessed 5. Feb. 2024].
The rest of the changes appears fine to me.
